# Molecular and Functional Characterization of Three General Odorant-Binding Protein 2 Genes in *Cydia pomonella* (Lepidoptera: Tortricidae)

**DOI:** 10.3390/ijms25031746

**Published:** 2024-02-01

**Authors:** Yanan Zhou, Cong Huang, Guanjun Fu, Rui Tang, Nianwan Yang, Wanxue Liu, Wanqiang Qian, Fanghao Wan

**Affiliations:** 1College of Plant Health & Medicine, Qingdao Agricultural University, Qingdao 266109, China; 2Shenzhen Branch, Guangdong Laboratory of Lingnan Modern Agriculture, Genome Analysis Laboratory of the Ministry of Agriculture and Rural Affairs, Agricultural Genomics Institute at Shenzhen, Chinese Academy of Agricultural Sciences, Shenzhen 518120, China; 3State Key Laboratory for Biology of Plant Diseases and Insect Pests, Institute of Plant Protection, Chinese Academy of Agricultural Sciences, Beijing 100193, China; 4Centre for Resource Insects and Biotechnology, Institute of Zoology, Guangdong Academy of Sciences, Guangzhou 510220, China

**Keywords:** *Cydia pomonella*, GOBPs, protein purification, fluorescence competitive binding assay, gene duplication

## Abstract

General odorant-binding proteins (GOBPs) play a crucial role in the detection of host plant volatiles and pheromones by lepidopterans. Previous studies identified two duplications in the *GOBP2* gene in *Cydia pomonella*. In this study, we employed qRT-PCR, protein purification, and fluorescence competitive binding assays to investigate the functions of three *GOBP2* genes in *C. pomonella*. Our findings reveal that *CpomGOBP2a* and *CpomGOBP2b* are specifically highly expressed in antennae, while *CpomGOBP2c* exhibits high specific expression in wings, suggesting a potential divergence in their functions. Recombinant proteins of CpomGOBP2a, CpomGOBP2b, and CpomGOBP2c were successfully expressed and purified, enabling an in-depth exploration of their functions. Competitive binding assays with 20 host plant volatiles and the sex pheromone (codlemone) demonstrated that CpomGOBP2a exhibits strong binding to four compounds, namely butyl octanoate, ethyl (*2E*,*4Z*)-deca-2,4-dienoate (pear ester), codlemone, and geranylacetone, with corresponding dissolution constants (Ki) of 8.59993 μM, 9.14704 μM, 22.66298 μM, and 22.86923 μM, respectively. CpomGOBP2b showed specific binding to pear ester (Ki = 17.37481 μM), while CpomGOBP2c did not exhibit binding to any tested compounds. In conclusion, our results indicate a functional divergence among *CpomGOBP2a*, *CpomGOBP2b*, and *CpomGOBP2c*. These findings contribute valuable insights for the development of novel prevention and control technologies and enhance our understanding of the evolutionary mechanisms of olfactory genes in *C. pomonella*.

## 1. Introduction

Insects have acquired robust olfactory approaches over the course of long-term evolution, contributing to their behaviors, including host localization, mate location, and selection of egg-laying sites [1,2]. Insect olfaction is facilitated by several types of olfactory proteins, such as odorant-binding proteins (OBPs) [3], chemosensory proteins (CSPs) [4], odorant receptors (ORs) [5], ionotropic receptors (IRs) [6], sensory neuron membrane proteins (SNMPs) [7], gustatory receptors (GRs) [8], and odorant-degrading enzymes (ODEs) [9]. The OBPs recognize hydrophobic odor molecules and deliver them to ORs, which highlight the remarkable sensitivity of insect olfactory systems.

Insect OBPs are a group of small soluble proteins that are confirmed to bind and transport the hydrophobic molecules [10,11]. They play essential roles in insect behaviors and are mainly classified into pheromone-binding proteins (PBPs), antennal-binding proteins (ABPs), and general odorant-binding proteins (GOBPs) [12,13]. GOBPs can be further classified into two classes, GOBP1 and GOBP2. The main functions of GOBPs are the perception of host volatiles and the sex pheromone, which have been reported in various moths and butterflies, including *Spodoptera frugiperda* [14], *Athetis lepigone* [15], *Conogethes pinicolalis* [16], *Hyphantria cunea* [17], *Grapholita funebrana* [18], *Orthaga achatina* [19], *Papilio xuthus* [20], and so on. In addition, some GOBPs have binding abilities to insecticides, such as *A. lepigone* [21], *Spodoptera litura* [22], and *P. xuthus* [20].

The codling moth, *Cydia pomonella* (L.) (Lepidoptera: Tortricidae), is a major pest of apples and pears, as well as other seed and stone fruits, which causes huge economic losses globally. Host plant volatiles and sex pheromones have been well applied to control *C. pomonella*, due to the strong chemosensory ability of *C. pomonella*. Previous studies identified 85 odorant receptors (ORs) and 40 odorant-binding proteins (OBPs) in *C. pomonella* [23]. Notably, the *CpomOR3* and *CpomGOBP2* genes have been duplicated over the course of the long-term evolution in *C. pomonella*. Studies have confirmed that the copies of *CpomOR3* enhanced the capabilities of mate location and host plant recognition. Using genome and transcriptome analysis, scientists determined that the *CpomGOBP2* genes have been duplicated twice, named *CpomGOBP2a*, *CpomGOBP2b*, and *CpomGOBP2c* [24,25]. However, the functions of these three *CpomGOBP2* genes remained unclear.

To examine the functions of these three *CpomGOBP2* genes, in this study, we first cloned the CDS of the full-length *CpomGOBP2a*, *CpomGOBP2b*, and *CpomGOBP2c* in *C. pomonella*. Subsequently, qRT-PCR was employed to investigate their relative expression across different developmental stages and tissues. The recombinant proteins were purified using an *Escherichia coli* expression system for these three CpomGOBP2 proteins [26,27]. Finally, the recombinant proteins were utilized to estimate their binding affinities to host plant volatiles and sex pheromone components using competitive fluorescence binding assays.

## 2. Results

### 2.1. Cloning and Protein Purification of CpomGOBP2

Amplification of complete length *CpomGOBP2* using the first strand of cDNA from RNA reverse transcription as a template was conducted. The PCR amplification was conducted using full-length primers to acquire the full length of *CpomGOBP2a*, *CpomGOBP2b*, and *CpomGOBP2c*. The agarose gel electrophoresis assay exhibited a clear band of interest at approximately 500 bp. The sequencing results of the target bands were consistent with transcriptome findings. The *CpomGOBP2a*, *CpomGOBP2b*, and *CpomGOBP2c* were 480 bp, 507 bp, and 507 bp in length, respectively.

The multiple sequence alignment results of coding sequences showed that the identity between *CpomGOBP2a* and *CpomGOBP2b* was 73.75%, the identity between *CpomGOBP2b* and *CpomGOBP2c* was 71.43%, and the identity between *CpomGOBP2a* and *CpomGOBP2c* was 60.17% (Figure 1a). The identities of amino acid sequences were 71.25% (CpomGOBP2a vs. CpomGOBP2b), 51.19% (CpomGOBP2b vs. CpomGOBP2c), and 39.62% (CpomGOBP2a vs. CpomGOBP2c), respectively (Figure 1b).

We reconstructed the phylogenetic tree of 19 lepidopteran *GOBP2* genes by RAxML, the result showed that three *CpomGOBP2* genes were clustered together (Figure 2a). By reconciling the gene tree with the species tree and rooting using NOTUNG, we obtained the rooted gene tree. The result suggested that *CpomGOBP2a* is the ancestral gene of *CpomGOBP2b* and *CpomGOBP2c* (Figure 2b).

### 2.2. Protein Purification of CpomGOBP2

The expression of CpomGOBP2 was executed in *E. coli*, whereby the recombinant CpomGOBP2a was 160 amino acids in length with a molecular weight of 18.33 kDa and an isoelectric point (pI) of 5.78. The CpomGOBP2b was 169 amino acids in length, with a molecular weight of 19.43 kDa and a pI of 5.22. The CpomGOBP2c was 169 amino acids in length, with a molecular weight of 19.19 kDa and a pI of 4.91. The purity and integrity of these recombinant proteins were verified using sodium dodecyl sulfate-polyacrylamide gel electrophoresis (SDS-PAGE).

### 2.3. Cloning and Spatiotemporal Expression Profiles of CpomGOBP2 Genes

There was no statistically significant difference identified in the expression levels of *CpomGOBP2a* between male and female adults (Figure 3a). However, *CpomGOBP2b* had significantly higher expression in males compared to females (Figure 3b), as female adults displayed minimal expression of *CpomGOBP2c* (Figure 3c). Regarding tissue-specific expression patterns, both *CpomGOBP2a* and *CpomGOBP2b* exhibited maximal expression levels in the antennae (Figure 3d,e). In contrast, *CpomGOBP2c* exhibited its highest expression in the wings, with significantly higher expression observed in males compared to females (Figure 3f).

### 2.4. Expression and Purification of Three Recombinant CpomGOBP2 Proteins

The recombinant proteins were expressed alongside an N-terminal His-tag. SDS-PAGE examination uncovered that CpomGOBP2a, CpomGOBP2b, and CpomGOBP2c were expressed as inclusion bodies (Figure 4a). The observed banding patterns on the SDS-PAGE gel were aligned with the expected molecular weights of CpomGOBP2a, CpomGOBP2b, and CpomGOBP2c (Figure 4b). The result was further confirmed by Western blot (Figure 4c). Subsequently, 1.7 mg of CpomGOBP2a, 1.89 mg of CpomGOBP2b, and 1.2 mg of CpomGOBP2c were successfully isolated. These purified recombinant proteins were utilized in subsequent fluorescence binding experiments.

### 2.5. Fluorescence Competitive Binding Assay

The affinity of the protein to the ligand was examined using a fluorescence competitive binding assay. The dissociation constants of CpomGOBP2a, CpomGOBP2b, and CpomGOBP2c with 1-NPN were 12.63 mM, 27.50 mM, and 18.85 mM, respectively, which indicates that 1-NPN is a good reporter ligand for these three OBPs (Figure 5). The fluorescence competitive binding assay indicated that the binding ability of these three recombinant CpomGOBP2 proteins to 20 candidate semiochemicals exhibits divergence. There are four compounds with exceptional binding affinity to CpomGOBP2a, which decreased the fluorescence intensity to half of the initial value during the titration, with dissolution constants (Ki) of 22.86923 μM for geranylacetone, 22.66298 μM for codlemone, 8.59993 μM for butyl octanoate, and 9.14704 μM for ethyl (*2E*,*4Z*)-deca-2,4-dienoate (Figure 6a). For CpomGOBP2b, only ethyl (*2E*,*4Z*)-deca-2,4-dienoate (Ki = 17.37481 μM) triggered the fluorescence value to be reduced to half of the initial value (Figure 6b). The binding activity of CpomGOBP2c to the experimental ligands was the poorest, with no fluorescence values dropping to 50% of the initial fluorescence value. However, the fluorescence value of ethyl (*2E*,*4Z*)-deca-2,4-dienoate (Ki = 48.3448 μM), which had the best binding performance, dropped only to 55.35% (Figure 6c).

### 2.6. Homology Modeling and Molecular Docking

To further understand the molecular interactions between the three CpomGOBP2 proteins and their ligands, we constructed their 3D structure models using homology modeling and molecular docking with their ligands. All three CpomGOBP2 proteins have seven α-helices, and their structures were very similar except for the different amino acid residues (Figure 7a–c). The docking results showed that the binding cavities were composed of hydrophobic amino acid residues located in the α-helix structures (Figure 7d and Appendix A). There are some same key amino acid residues in the binding pockets of CpomGOBP2a and CpomGOBP2b with the same ligand ethyl (*2E*,*4Z*)-deca-2,4-dienoate, such as 24M, 27V, 56W, 71I, 80L, 92M, 95Y, and 137F, however, among these amino acid residues, there are several different residues in the CpomGOBP2c, including 24M, 27V, 92M, 95Y, and 137F (Figure 7d, Appendix A). For the butyl octanoate ligand, the key amino acid residues of CpomGOBP2a are composed of 27V, 28T, 31F, 52F, 55F, 56W, 71I, 80L, 92M, 95Y, 109M, 113I, 130V, 133V, 134A, and 137F; among them, 28T, 31F, 52F, 55F, 109M, 113I, and 134A were different from the CpomGOBP2b and CpomGOBP2c (Figure 7d, Appendix A). For the codlemone ligand, the key amino acid residues of CpomGOBP2a are composed of 28T, 52F, 55F, 56W, 71I, 81L, 85A, 86R, 87M, 113I, 117E, 129R, 130V, 133V, 134A, and 137F; among them, 28T, 52F, 55F, 85A, 113I, and 134A are different from the CpomGOBP2b and CpomGOBP2c (Figure 7d, Appendix A). For the geranylacetone ligand, the key amino acid residues of CpomGOBP2a are composed of 24M, 27V, 28T, 31F, 52F, 55F, 56W, 71I, 75S, 80L, 92M, 95Y, 109M, 113I, 134A, and 137F; among them, 28T, 31F, 52F, 55F, 109M, 113I, and 134A are different from the CpomGOBP2b and CpomGOBP2c (Figure 7d, Appendix A).

## 3. Discussion

The general odorant-binding proteins have critical functions in the recognition of host plants and mates in lepidopteran insects [15,28]. In previous studies, a total of four general odorant-binding protein genes were characterized in *C. pomonella*, including one GOBP1 and three duplicates of the GOBP2 gene [24,25]. The three GOBP2 genes were named *CpomGOBP2a*, *CpomGOBP2b*, and *CpomGOBP2c*. To investigate the functions of the three GOBP2 genes in *C. pomonella*, we cloned their full-length CDS. The results indicated that the CDS lengths of *CpomGOBP2a*, *CpomGOBP2b*, and *CpomGOBP2c* were 480 bp, 507 bp, and 507 bp, with corresponding amino acid lengths of 160, 169, and 169, respectively. The sequence length was consistent with previous results [29,30]. According to the phylogenetic tree, we inferred that *CpomGOBP2a* is the ancestral gene of *CpomGOBP2b* and *CpomGOBP2c*.

The mRNA expression profiles across different stages and tissues indicated that *CpomGOBP2a* was specifically highly expressed in adults and the antennae of both sexes and the expression levels were similar across both sexes. This result is similar to the results of the GOBP1 genes in *Spodoptera exigua* [29] and *Conopomorpha sinensis* [31], which suggests that *CpomGOBP2a* is involved in chemosensory. The expression profiles of *CpomGOBP2b* and *CpomGOBP2c* were similar; both were highly expressed in male adults, followed by the 5th male or female larva. The expression patterns were similar to *DabiGOBP2* of *Dioryctria abietella* [30] and *BdorOBP21* of *Bactrocera dorsalis* [32], suggesting that they are involved in the olfaction of both adults and larvae. In addition, *CpomGOBP2b* was mainly highly expressed in males, indicating that it may play important functions in males. Similar results were observed in other lepidopteran species, including *GmolGOBP1* of *Grapholita molesta* and *DtabGOBP1* of *Dendrolimus tabulaeformis,* where its expression in male antennae was more elevated than in female antennae [33,34]. The transcript levels of GOBP2 genes from *Conopomorpha sinensis* [31], *Plutella xylostella* [35], and *Loxostege sticticalis* [36] in male moths were higher than those in female moths. Previous studies indicated that the OBPs highly expressed in wings may be involved in gustatory functions, such as the *BodoOBP17*, *BodoOBP30*, *BodoOBP32*, *BodoOBP37*, and *BodoOBP44* in *Bradysia odoriphaga*, which were abundantly expressed in the wings. The author suggested that these genes might also participate in taste functions [37]. In this study, *CpomGOBP2c* was highly expressed in the wings of adults, suggesting that it not only plays an important role in olfaction but also may be involved in taste functions. Considering the characteristics of expression profiles, we infer that the functions of the three *CpomGOBP2* genes have diverged.

In lepidopteran insects, each species has one *GOBP1* and one *GOBP2* gene. However, as more GOBP genes were identified, the GOBP genes have expanded or contracted throughout several species. A prior study determined that the duplication and subsequent translocation of the *GOBP1* gene took place in both *P. xylostella* and *Operophtera brumata* [38]. Generally, there are three fates of duplicated genes: they can (1) degenerate to a pseudogene or non-functionalization, (2) enhance the gene functions, and (3) generate novel gene functions [39]. A previous study identified that the duplicates of the *CpomOR3* gene enhanced the functions of host and mate recognition in *C. pomonella* [23]. In *Drosophila melanogaster*, the *DmelOR22b*, a duplicate of the *OR22* gene, became nonfunctional, while *DmelOR22ab* changed its ligand binding ability [40]. In our previous study, we determined that the *GOBP2* genes were duplicated two times, which were named *CpomGOBP2a*, *CpomGOBP2b*, and *CpomGOBP2c*. To characterize the functions of the three *GOBP2* genes, we cloned their full-lengths and purified the proteins. Subsequently, we selected 20 compound ligands that could represent the volatiles of host plants or other interaction targets in *C. pomonella*. Among them, the dodecane, tridecane, tetradecane, Δ-cadinene, 2-octanone, nonanal, butyl octanoate, cis-3-Hexen-1-ol, α-farnesene, 2-cyclopentylcyclopentanone, ethyl (*2E*,*4Z*)-deca-2,4-dienoate, DMNT, β-caryophyllene, and Linalool were the volatiles of host plants [41,42,43], and the sulcatone, geranylacetone, 2-phenylethanol, 3-methyl-3-butenol, and 2-phenylethyl acetate were the volatiles of yeast [44]. Codlemone is the major sex pheromone compound [45]. Competitive binding assays found that the three gene functions have undergone divergence. CpomGOBP2a has a broad binding ability, possibly binding to four compounds, including geranylacetone, codlemone, butyl octanoate, and ethyl (*2E*,*4Z*)-deca-2,4-dienoate, and the result is similar to other studies, such as the AlepGOBP2 of *A. lepigone* [15], the SfruGOBP2 of *S. frugiperda* [14], the OachGOBP2 of *O. achatina* [19], and the HcunGOBP2 of *H. cunea* [17], which have a broader ligand-binding spectrum. In contrast, CpomGOBP2b could only bind to pear ester, and CpomGOBP2c was unable to bind to any compounds. To explore the reason for functional differentiation, we performed molecular docking; the results showed that in the ligand binding pockets, there are some key amino acid residues of CpomGOBP2a that are different from CpomGOBP2b and CpomGOBP2c, which may contribute to their differentiated functions. Our results enriched the view of the functional evolution of duplicate genes.

In conclusion, according to the previous findings that the *CpomGOBP2* gene has been duplicated two times, our current study first cloned the full-length of the *CpomGOBP2a*, *CpomGOBP2b*, and *CpomGOBP2c* genes. The developmental stages and tissue expression profile analyses indicated that the functions of the three *CpomGOBP2* genes may have diverged to generate novel functions. The competitive binding assay identified the ligands of the three *CpomGOBP2* genes and verified their functional differentiation. The differences in some key amino acid residues in ligand binding pockets may have led to their functional differentiation. Our findings provide a foundation for the development of new prevention and control technologies and enhance the study of evolutionary mechanisms of olfactory genes in *C. pomonella*.

## 4. Materials and Methods

### 4.1. Insect Rearing and Tissue Collection

The *C. pomonella* were reared at 26 ± 2 °C, with a relative humidity of 60 ± 5% and a 16 h light:8 h darkness photoperiod. Various development samples (including eggs, larvae, pupae, and adults) and different tissue samples (including antennae, heads, thoraces, abdomens, legs, and wings) were dissected and immediately transferred to RNase-free tubes. The samples were flash-frozen using liquid nitrogen and stored at −80 °C until RNA extraction.

### 4.2. Total RNA Extraction and cDNA Cloning

Total RNA was extracted using a small volume total RNA extraction kit (Jianshi Biotech, Beijing, China). The purity of the RNA samples was examined using absorbance ratios of A260/A280 and A260/A230, the integrity was verified by electrophoresis on 1.0% agarose gel, and the concentration was identified using a Nanodrop One (Thermo Fisher Scientific, Waltham, MA, USA). cDNA libraries were prepared using Hifair III 1st Strand cDNA Synthesis SuperMix for qPCR (Yeasen, Shanghai, China), following the manufacturer’s instructions.

### 4.3. Multiple Sequence Alignment and Phylogenetic Analysis

Multiple sequence alignment (MSA) of the coding sequences and amino acid sequences of *CpomGOBP2a*, *CpomGOBP2b*, and *CpomGOBP2c* were performed using MAFFT v7 [46] with default parameters. Those alignment results were visualized using GeneDoc [47]. To confirm the sequence identities of these three genes, we submitted their coding sequences and amino acid sequences to the EBI Clustal Omega Server (https://www.ebi.ac.uk/Tools/msa/clustalo/; accessed on 2 October 2023) to perform multiple sequence alignment with a global alignment method.

To explore the evolutionary relationship of the lepidopteran general binding protein 2, we downloaded the GOBP2 amino acid sequences of 16 lepidopteran insects from NCBI, which included *Spodoptera frugiperda* (WKR38877.1), *Spodoptera litura* (AKI87961.1), *Amyelois transitella* (XP_013195990.2), *Manduca sexta* (XP_030025611.1), *Bombyx mori* (NP_001037498.1), *Plutella xylostella* (CAG9127646.1), *Helicoverpa armigera* (PZC76187.1), *Trichoplusia ni* (XP_026739044.1), *Danaus plexippus* (XP_032519593.2), *Heliconius melpomene* [48], *Pieris rapae* (QNS26338.1), *Papilio xuthus* (KPJ00902.1), *Papilio polytes* (XP_013136922.1), *Papilio machaon* (XP_014368562.2), *Melitaea cinxia* (XP_045452652.1), and *Spodoptera exigua* (CAD0249197.1). A total of 19 GOBP2 amino acid sequences were performed with multiple sequence alignment by using MAFFT v7 software with default parameters. Afterward, the aligned sequences were trimmed by trimAl v1.2 [49] to remove gaps and low-quality regions with the parameter “-automated1”. The phylogenetic tree with 1000 bootstrap replicates was inferred using maximum likelihood (ML) in RAxML version 8.2.12 [50], with the best-fit model (JTT) estimated by ProtTest3 v3.4.2 [51]. This gene tree and the species tree that was built in a previous study [52] were then reconciled using the NOTUNG software (Version 2.9) [53], and a root was placed on the GOBP2 gene tree using the rooting procedure in NOTUNG [53]. All the phylogenetic trees were visualized with FigTree v1.4.3 (http://tree.bio.ed.ac.uk/software/figtree/; accessed on 6 October 2023).

### 4.4. Expression and Purification of Three CpomGOBP2 Proteins

The proteins were recombinantly expressed using an Escherichia coli expression system with an N-His tag. The complete coding regions lacking the signal peptide sequence of CpomGOBP2a, CpomGOBP2b, and CpomGOB2c were subcloned into the NdeI/EcoRI restricted and dephosphorylated pET-28b expression vector. The constructed plasmid was subsequently transformed into *E. coli* BL21 (DE3) competent cells for further expression. The size and purity of the target proteins were confirmed using sodium dodecyl sulfate-polyacrylamide gel electrophoresis (SDS-PAGE) and Western blot. The proteins were stored at −80 °C for subsequent fluorescence competitive binding experiments.

### 4.5. Expression Profiles of Three CpomGOBP2 Genes

The qRT-PCR was conducted using SYBR Green Master Mix (Yeasen Biotech, Shanghai, China) on an Applied Biosystems 7500 (Thermo Fisher Scientific, Waltham, MA, USA) real-time quantitative fluorescence PCR system. Each reaction was performed using a total volume of 20 µL, containing 1 µL of template cDNA, 10 µL of SYBR Green Master Mix, 0.4 µL of each primer (10 µM), and 8.2 µL of ddH_2_O. The PCR was conducted using a program of 30 s at 95 °C, followed by 40 cycles of 30 s at 95 °C and 34 s at 60 °C. A melting curve was constructed at 95 °C for 60 s. The amplification efficiency of each gene was optimized to maximize the peak value of each gene throughout the amplification process. The stable expression of the elongation factor 1-α (*EF-1α*) gene in *C. pomonella* was identified as an internal reference gene to correct the true expression levels of *CpomGOBP2* genes in different samples. The relative gene expression was calculated using the 2^−ΔΔCT^ method. The results were reported as the means (*n* = 3) ± standard errors; significant differences in expression levels were determined using an independent samples *t*-test and plotted using GraphPad Prism 8.0.

### 4.6. Fluorescence Competitive Binding Assay

Fluorescence competitive binding assays were conducted on a Synergy4 (BioTek Instruments, Winooski, VT, USA) microplate reader using the F96 Black ELIAS Plate (Xinyou Biotechnology, Hangzhou, China). Probe 1-NPN and the tested ligands were dissolved in spectrophotometric-grade methanol to generate 1.0 mM stock solutions. The fluorescence probe 1-NPN was excited at 337 nm, and emission spectra were recorded between 390 and 490 nm. Initially, to test the binding constants of 1-NPN to the CpomGOBP2, a 2.0 µM solution of protein in 50 mM Tris-HCl (pH = 7.4) was titrated with 1 mM of 1-NPN to achieve various concentrations. Next, the competitive binding of each odorant (Table 1) was examined using 1-NPN as a fluorescent reporter and the odorant as a competitor. The concentrations of protein and 1-NPN were both 2.0 µM; the odorant was included after the protein and 1-NPN were added into the well of the ELIAS Plate for 2 min. The final concentrations of each competitor were 2, 4, 6, 8, 10, 12, 14, 16, 18, and 20 µM. After 2 min of odorant addition, the fluorescence intensity was measured and documented. The volume of mixed solution in each well was maintained at 250 µL. Each interaction was conducted in triplicate.

### 4.7. Modeling and Molecular Docking

To further explore the reason for the functional differentiation in the two *CpomGOBP2* genes, *CpomGOBP2b* and *CpomGOBP2c*, compared to their ancestral gene, *CpomGOBP2a*, the amino acid sequences of the three *CpomGOBP2* genes were submitted to the SWISS-MODEL Server5 to predict and refine the 3D structures. All the 3D structures were built by referring to the best template, BmorGOBP2 (PDB ID: 2WCK). Subsequently, the web server, SAVES server6 (https://saves.mbi.ucla.edu/; accessed on 7 October 2023), was used to estimate the quality of the predicted 3D structures. The generated model structures were rendered and visualized using PyMOL v2.2.3 [54]. Then, we performed molecular docking for the two CpomGOBP2 proteins and their ligands in Molecular Operating Environment (MOE). The binding patterns between the two CpomGOBP2 proteins and their odorant molecule ligands, as well as the key binding sites, were visualized using MOE. The detailed steps of molecular docking were described in the previous study [55].

## Figures and Tables

**Figure 1 ijms-25-01746-f001:**
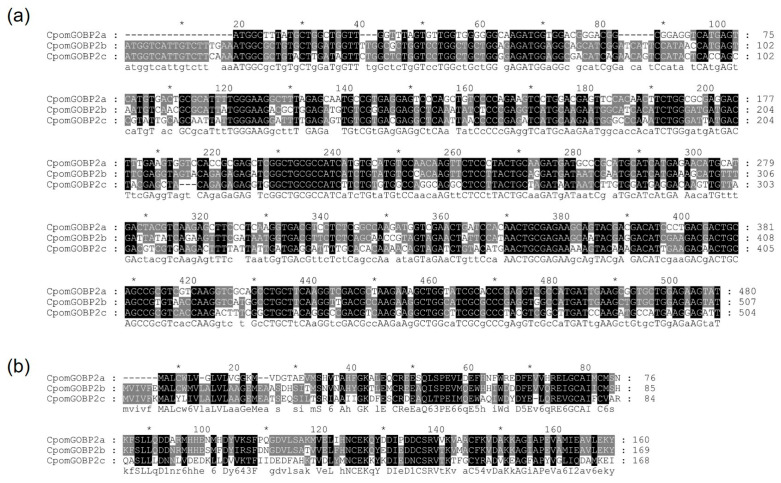
Multiple sequence alignment of the three *CpomGOBP2* genes. (**a**) Multiple sequence alignment of the coding sequences of three *CpomGOBP2* genes. (**b**) Multiple sequence alignment of the amino acid sequences of three *CpomGOBP2* genes. The grey shading indicating the nucleobase or amino acid identity. The * indicates the position of the nucleobase or amino acid residues between the preceding and following numbers.

**Figure 2 ijms-25-01746-f002:**
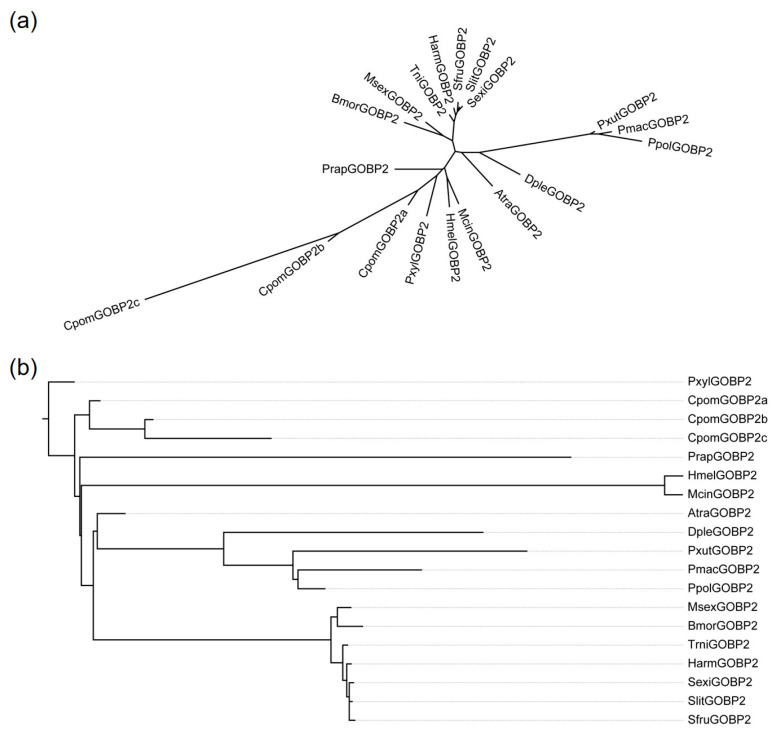
Phylogenetic tree of 19 *GOBP2* genes from 17 lepidopteran insects. (**a**) Maximum-likelihood tree of 19 *GOBP2* genes constructed by RAxML. (**b**) A rooted-species-reconciled phylogenetic gene tree by reconciling the gene tree with the species tree and rooting using NOTUNG. Pxyl (*Plutella xylostella*), Cpom (*Cydia pomonella*), Prap (*Pieris rapae*), Hmel (*Heliconius melpomene*), Mcin (*Melitaea cinxia*), Sfru (*Spodoptera frugiperda*), Slit (*Spodoptera litura*), Atra (*Amyelois transitella*), Msex (*Manduca sexta*), Bmor (*Bombyx mori*), Harm (*Helicoverpa armigera*), Trni (*Trichoplusia ni*), Dple (*Danaus plexippus*), Pxut (*Papilio xuthus*), Ppol (*Papilio polytes*), Pmac (*Papilio machaon*), Sexi (*Spodoptera exigua*).

**Figure 3 ijms-25-01746-f003:**
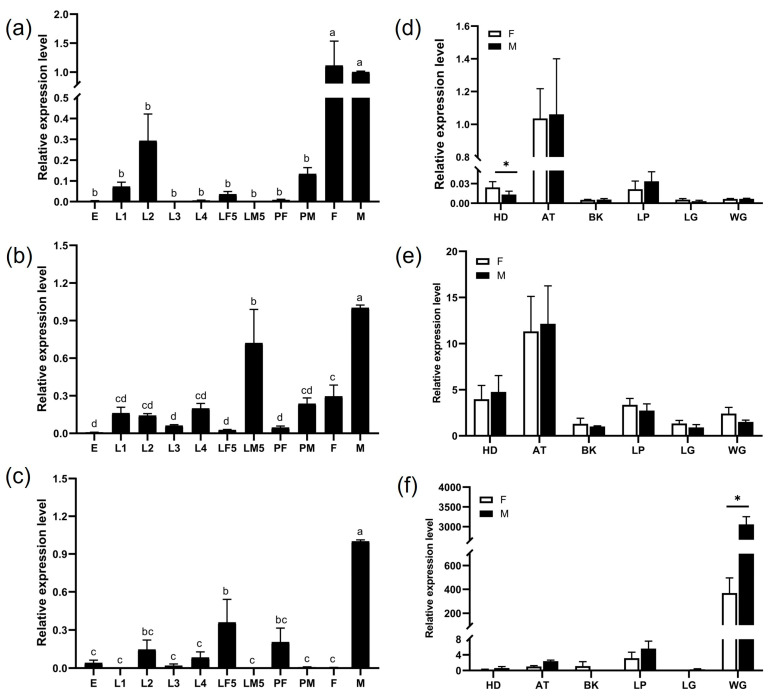
Relative mRNA expression of *CpomGOBP2* in *Cydia pomonella*. (**a**) Tissue expression profile of *CpomGOBP2a*; (**b**) tissue expression profile of *CpomGOBP2b*; (**c**) tissue expression profile of *CpomGOBP2c*; (**d**) spatiotemporal expression profile of *CpomGOBP2a*; (**e**) spatiotemporal expression profile of *CpomGOBP2b*; (**f**) spatiotemporal expression profile of *CpomGOBP2c*. E: egg, L1: 1st instar larvae, L2: 2nd instar larvae, L3: 3rd instar larvae, L4: 4th instar larvae, LF5: 5th instar female larvae, LM5: 5th instar male larvae, PF: female pupa, PM: male pupa, F: female adult, M: male adult, HD: head, AT: antenna, BK: beak, LP: labipalp; LG: leg, WG: wing. All values are reported as the mean + SEM and normalized. Different letters mean the significant difference between different treated proteins (*p* < 0.05, ANOVA, Tukey’s HSD). * *p* < 0.05.

**Figure 4 ijms-25-01746-f004:**
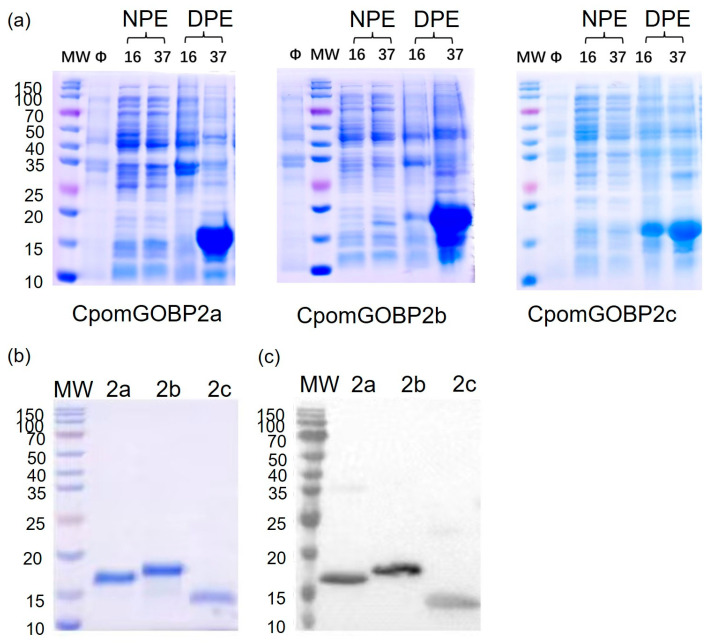
Expression and purification of three CpomGOBP2 proteins: (**a**) the recombinant proteins were identified as being present in inclusion bodies. Protein molecular weight marker (M), from the top: 150, 100, 70, 50, 40, 35, 25, 20, 15, 10 kDa. NPE: supernatant, DPE: inclusion bodies, Ø: negative control. (**b**) The purification of CpomGOBP2 proteins. (**c**) The Western blot of CpomGOBP2 proteins.

**Figure 5 ijms-25-01746-f005:**
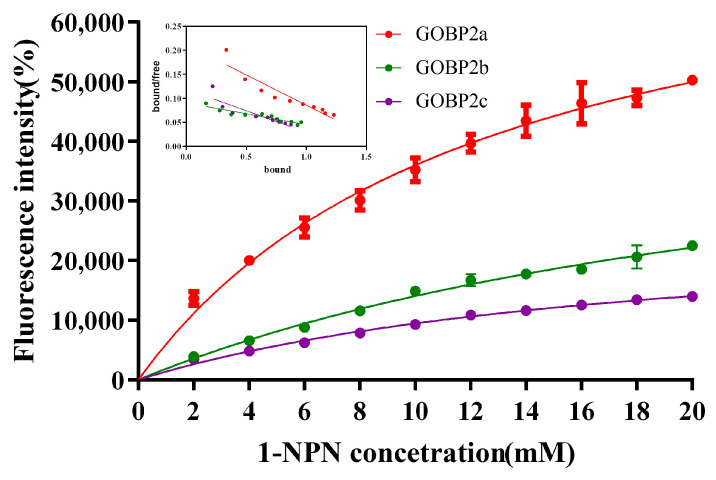
Binding curves and Scatchard plots (insert) of 1-NPN to CpomGOBP2a, CpomGOBP2b, and CpomGOBP2c.

**Figure 6 ijms-25-01746-f006:**
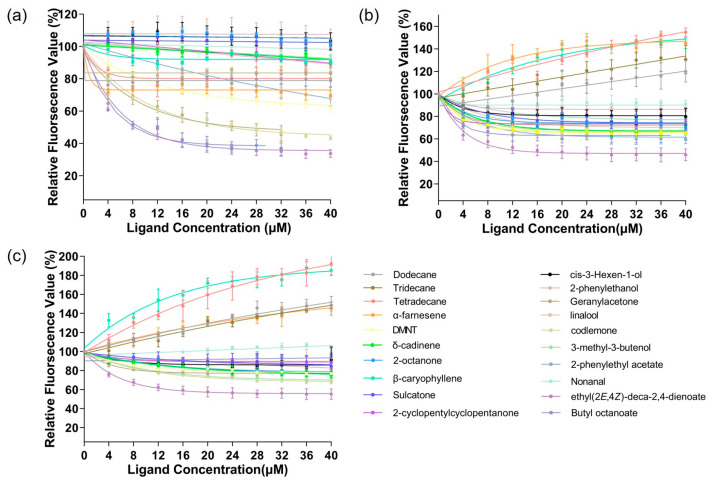
Competitive binding curves for CpomGOBP2a, CpomGOBP2b, and CpomGOBP2c toward investigated ligands. (**a**): CpomGOBP2a. (**b**): CpomGOBP2b. (**c**): CpomGOBP2c.

**Figure 7 ijms-25-01746-f007:**
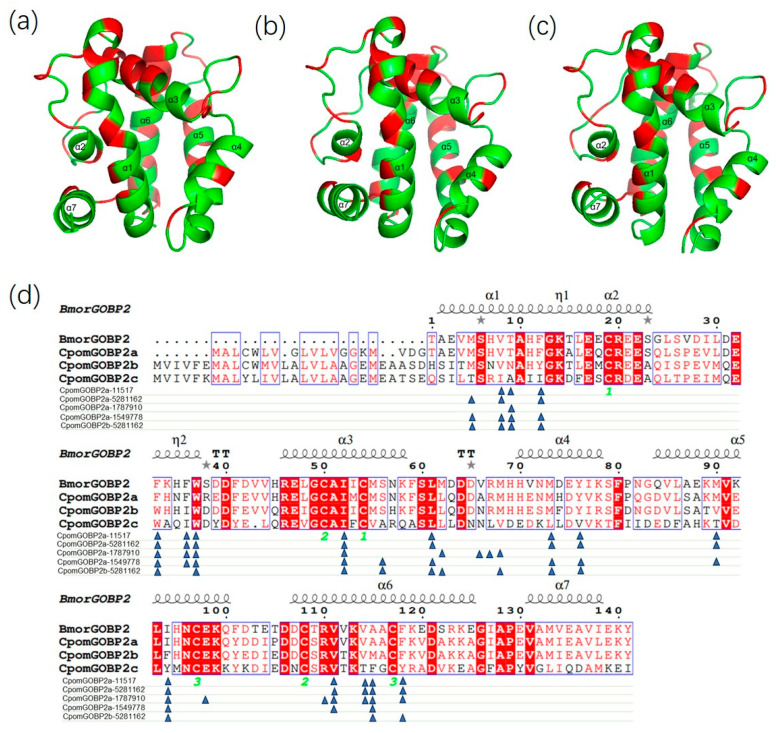
3D structures of the three CpomGOBP2 proteins and their key amino acid sites interacting with their ligands. (**a**) CpomGOBP2a. (**b**) CpomGOBP2b. (**c**) CpomGOBP2c. The α1-α7 are α-helixes. The red sites in the structures are the conserved sites of the three CpomGOBP2 genes. The green sites in the structures are not conserved. (**d**) Key amino acid sites of CpomGOBP2a and CpomGOBP2b with their ligands. The blue triangles show the interaction sites. The green numbers “1”, “2” and “3” indicated the cysteine residues forming the three disulfide bonds. The loop with alternative conformations is indicated with grey stars above. The ligands are represented by the CID number: 11517 represents butyl octanoate, 5281162 represents ethyl (*2E*,*4Z*)-deca-2,4-dienoate (pear ester), 1787910 represents codlemone, and 1549778 represents geranylaceton.

**Table 1 ijms-25-01746-t001:** Summary of 20 odor samples for the fluorescence competitive binding experiment.

Ligands	CAS Number	Molecular Weight	Purity (%)
Dodecane	112-40-3	170.335	98
Tridecane	629-50-5	184.361	98
Tetradecane	629-59-4	198.388	99
α-farnesene	502-61-4	204.351	98
DMNT	19945-61-0	150.261	94
Δ-cadinene	7705-14-8	136.234	95
β-caryophyllene	87-44-5	204.351	90
2-octanone	111-13-7	128.212	98
Sulcatone	110-93-0	126.196	98
2-cyclopentylcyclopentanone	4884-24-6	152.233	97
Cis-3-Hexen-1-ol	928-96-1	100.159	98
Geranylacetone	3796-70-1	194.313	98
2-phenylethanol	60-12-8	122.164	99
Linalool	78-70-6	154.25	98
Codlemone	33956-49-9	182.302	90
3-methyl-3-butenol	763-32-6	86.132	98
Nonanal	124-19-6	142.24	99
2-phenylethyl acetate	103-45-7	164.201	98
Butyl octanoate	589-75-3	200.318	99
Ethyl (*2E*,*4Z*)-deca-2,4-dienoate	3025-30-7	196.286	97

## Data Availability

The data that support the findings of this study are available from the corresponding author upon reasonable request.

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
