# Peer review of "Molecular and Functional Characterization of Three General Odorant-Binding Protein 2 Genes in Cydia pomonella (Lepidoptera: Tortricidae)"

_ijms, 2024, doi:10.3390/ijms25031746_

Round 1
Reviewer 1 Report
Comments and Suggestions for Authors
GOBP genes in insects show a relatively conserved nature, and many studies have employed purified GOBP to study their functions, as shown in this study. However, the authors have omitted a summary of these studies in the introduction section. It is recommended that the authors familiarize themselves with relevant literature, particularly papers related to odorant binding proteins in Cydia pomonella. An example is the work by Huang et al., titled "Comparative genomics provide insights into function and evolution of odorant binding proteins in Cydia pomonella".
Some extensively studied GOBPs were overlooked in this study, and the rationale behind investigating the function of GOBP2s in Cydia pomonella remains unclear. Recent studies, such as "The binding affinity of two general odorant binding proteins in Spodoptera frugiperda to general volatiles and insecticides," conducted sequence alignments of SfruGOBP1 and SfruGOBP2, along with constructing a phylogenetic tree of GOBPs from Lepidopteran insects. In contrast, this manuscript lacks displays of conserved domains and phylogenetic trees of GOBPs, as seen in the paper.
For a comprehensive study of GOBP function in Cydia pomonella, it is a good way to extract all OBP family members from the Cydia pomonella genome and localize GOBP2 in a phylogenetic tree of OBPs. Additionally, to enhance the study's depth, it is suggested to expand the range of compounds tested for binding activity of GOBP2s, to uncover similarities and specific activities when compared to GOBPs in other insects.
The study didn’t confirm the purity of the isolated GOBP2s through Western blots or Mass spectrometry. And the lane names in fig 2 are not correct, please double check them. Addressing these issues and considering the novelty of the study would significantly contribute to its overall quality.
Comments on the Quality of English Language
Quality of English Language is OK.
Author Response
|
Comments 1: GOBP genes in insects show a relatively conserved nature, and many studies have employed purified GOBP to study their functions, as shown in this study. However, the authors have omitted a summary of these studies in the introduction section. It is recommended that the authors familiarize themselves with relevant literature, particularly papers related to odorant binding proteins in Cydia pomonella. An example is the work by Huang et al., titled "Comparative genomics provide insights into function and evolution of odorant binding proteins in Cydia pomonella". |
|
Response 1: Thanks for your suggestion. We added some introductions on the functions of insect GOBPs, especially for the studies on the codling moth. As seen in line 45-54. |
|
Comments 2: Some extensively studied GOBPs were overlooked in this study, and the rationale behind investigating the function of GOBP2s in Cydia pomonella remains unclear. Recent studies, such as "The binding affinity of two general odorant binding proteins in Spodoptera frugiperda to general volatiles and insecticides," conducted sequence alignments of SfruGOBP1 and SfruGOBP2, along with constructing a phylogenetic tree of GOBPs from Lepidopteran insects. In contrast, this manuscript lacks displays of conserved domains and phylogenetic trees of GOBPs, as seen in the paper. |
|
Response 2: We added the multiple sequence alignment of three CpomGOBP2 genes, including the coding sequences and the amino acid sequences to analysis the sequence differences among the three CpomGOBP2, as well as their conserved domains. To study the evolutionary relationship of the three CpomGOBP2 genes, we constructed a phylogenetic tree of 19 Lepidopteran GOBP2 genes from 17 insects. And re-rooted the GOBP2 gene tree to explore the evolutionary status of three CpomGOBP2 genes, the result suggests that the CpomGOBP2a is the ancestral gene of CpomGOBP2b and CpomGOBP2c. In addition, we also built the 3D structures for the three CpomGOBP2 proteins and performed the molecular docking to explore the conserved domains and active binding sites. As seen in Figure 1 and Figure 2. |
|
Comments 3: For a comprehensive study of GOBP function in Cydia pomonella, it is a good way to extract all OBP family members from the Cydia pomonella genome and localize GOBP2 in a phylogenetic tree of OBPs. Additionally, to enhance the study's depth, it is suggested to expand the range of compounds tested for binding activity of GOBP2s, to uncover similarities and specific activities when compared to GOBPs in other insects. |
|
Response 3: In our previous study, we have identified all the OBP family members of C. pomonella and constructed the phylogenetic tree of OBPs (Huang et al., 2021, doi: 10.3389/fphys.2021.690185), in this study, our main target is exploring the functions of three CpomGOBP2 genes, so we didn’t rebuild the phylogenetic tree. Instead, we constructed a phylogenetic tree of 19 lepidopteran GOBP2 genes from 17 insects to explore the evolutionary status of three CpomGOBP2 genes. We selected some representative compounds to explore if the duplicates of CpomGOBP2 enhance the function of CpomGOBP2 or there is a functional differentiation. So, we think the ligand panel has a representativeness. We added the literatures of the selected compounds. Our results suggested that the functions of three CpomGOBP2 genes have differentiation. As seen in line 242-249. |
|
Comments 4: The study didn’t confirm the purity of the isolated GOBP2s through Western blots or Mass spectrometry. And the lane names in fig 2 are not correct, please double check them. Addressing these issues and considering the novelty of the study would significantly contribute to its overall quality. |
|
Response 4: It is a little insufficient without Western blots or Mass spectrometry. Before the protein purification, we have confirmed the accuracy and uniqueness for each CpomGOBP2 genes by sequencing. In addition, we used the single gene expression system, so we think about that the SDS-PAGE could further confirm each isolated CpomGOBP2. We have corrected the lane name in Figure 4. |

Reviewer 2 Report
Comments and Suggestions for Authors
The manuscript by Zhou et al describes three variants of an odorant binding protein, GOBP2a, in the codling moth, describing differences in where the three genes are expressed, and differences in the proteins’ binding affinities to a suite of odorants. The study builds on a previous one that identified the 3 variant genes: GOBP2a, b, and c, providing evidence that the genes and their associated proteins have diverged in predicted functions. The study would be of interest to researchers with interests in OBP evolution and those interested in developing new approaches to this pest insect’s control using odorant baits or disruptors.
The study is quite focused in its findings, and I think it would benefit from some elaboration on the following points before publication:
1. What are the sequence differences of the three GOBP2 genes? These should be summarized in this manuscript, rather than asking readers to seek the previous study to know about the gene’s differences. In particular, is anything known of the predicted ligand binding domains? Ligand binding pockets have been characterized in other species, so could these proteins be overlaid onto those structures to examine the differences among the three proteins? Do they show few or many amino acid substitutions in those regions that may account for the differences in odorant binding? Given that GOBP2c failed to bind the tested odorants, does the predicted binding domain show major changes relative to the other two?
2. How were the 20 odorants selected? What is known about whether these odorants are strong attractants for the moths, and do males and females display differences in their responses to them? Some more details on the relevance of these odorants could have been provided in the Discussion.
3. The bacterially-expressed proteins were able to bind to some odorants, but do the authors think that the bacterially expressed proteins could be post-translationally modified if they were expressed in a eukaryotic system? Can the authors provide a rationale for choosing E. coli rather than yeast, insect, or mammalian cells?
4. The authors suggested that their findings could be informative in studies of olfactory gene evolution. Do any other species share duplications of the GOBP2 gene, and if so, what comparisons can be made?
5. The GOBP2a protein appears to have broad binding activity, binding at least 4 odorants. Do those compounds share chemistries/structures that may explain their ability to bind effectively to this OBP?
6. Can the authors provide some comparisons to odorant binding proteins in other insects, such as OBPs’ abilities to bind multiple ligands?
7. Figure 1’s caption is missing an explanation of the abbreviations in panels d-f.
Comments on the Quality of English LanguageI would recommend checking sentence structure using a program such as Grammarly, as there are several grammatical errors that were overlooked.
Author Response
|
Comments 1: What are the sequence differences of the three GOBP2 genes? These should be summarized in this manuscript, rather than asking readers to seek the previous study to know about the gene’s differences. In particular, is anything known of the predicted ligand binding domains? Ligand binding pockets have been characterized in other species, so could these proteins be overlaid onto those structures to examine the differences among the three proteins? Do they show few or many amino acid substitutions in those regions that may account for the differences in odorant binding? Given that GOBP2c failed to bind the tested odorants, does the predicted binding domain show major changes relative to the other two? |
|
Response 1: Thanks for your suggestions, we have added the multiple sequence alignment of three CpomGOBP2 genes to analysis their identities between each other. We also built the 3D structures for the three CpomGOBP2 proteins and performed the molecular docking to explore the conserved domains and active binding sites. The docking results suggested that in the ligand binding pockets, there are some key amino acid residues of CpomGOBP2a that are different with CpomGOBP2b and CpomGOBP2c, which may contribute to their differentiated functions. As see in Figure 1 and Figure S1-S10. |
|
Comments 2: How were the 20 odorants selected? What is known about whether these odorants are strong attractants for the moths, and do males and females display differences in their responses to them? Some more details on the relevance of these odorants could have been provided in the Discussion. |
|
Response 2: We added the references of the 20 selected odorants and detailed their functions in the discussion section. As see in line 242-249. |
|
Comments 3: The bacterially-expressed proteins were able to bind to some odorants, but do the authors think that the bacterially expressed proteins could be post-translationally modified if they were expressed in a eukaryotic system? Can the authors provide a rationale for choosing E. coli rather than yeast, insect, or mammalian cells? |
|
Response 3: It is a very good question, we couldn’t find some reports about that the post-translationally modified occur in odorant binding proteins, we think the future research should focus on this. We selected the E. coli expression system like many other studies of the OBP functions because of its low cost, high expression capacity, easy purification of expression products, good stability, strong resistance to contamination, and a wide range of applications (Zhu et al., 2017, doi:10.1073/pnas.1711437114; Wang et al., 2022, doi.org/10.1016/j.cub.2021.12.054). |
|
Comments 4: The authors suggested that their findings could be informative in studies of olfactory gene evolution. Do any other species share duplications of the GOBP2 gene, and if so, what comparisons can be made? |
|
Response 4: By mining the related literatures, we couldn’t find other species share the duplications of the GOBP2 gene, however, there are some species have the duplications of the GOBP1 gene, such as a previous study determined that the duplication and subsequent translocation of the GOBP1 gene took place in both Plutella xylostella and Operophtera brumata, we discussed it in the discussion section. In addition, the duplication of olfactory genes in insects is a relatively common phenomenon, such as the duplication of CpomOR3 gene in Cydia pomonella (Wan et al., 2019, doi.org/10.1038/s41467-019-12175-9) and the duplication of OR5 gene in an ancestor of Spodoptera littoralis and Spodoptera litura (Li et al., 2023, doi.org/10.1073/pnas.2221166120). |
|
Comments 5: The GOBP2a protein appears to have broad binding activity, binding at least 4 odorants. Do those compounds share chemistries/structures that may explain their ability to bind effectively to this OBP? |
|
Response 5: We couldn’t find obvious similarity in chemistries/structures of those compounds, however, we performed the molecular docking for the CpomGOBP2 proteins with their ligands, the results suggested that the broad binding activity may be related to its key amino acid residues, and we also found many same cases in other species that their GOBP2 proteins have the broad binding activity with diversity ligands, and added this in discussion section. As see in line 256-259, Figure S1-S10. |
|
Comments 6: Can the authors provide some comparisons to odorant binding proteins in other insects, such as OBPs’ abilities to bind multiple ligands? |
|
Response 6: Thanks for your suggestions, we have added some similar cases that the GOBP2 proteins from other species have the broad binding activity with multiple ligands in the discussion section. As see in line 252-255. |

Round 2
Reviewer 1 Report
Comments and Suggestions for Authors
I recommend that identifying the purified OBP using WB or Mass spectrum is essential since all functional assays rely on the use of purified protein.
Author Response
Comments : I recommend that identifying the purified OBP using WB or Mass spectrum is essential since all functional assays rely on the use of purified protein.
Response : Thanks for your suggestion. We added western blot to identify the purified CpomGOBP2. As see in Figure 4c.